# Using Artificial Intelligence to Analyze Non-Human Drawings: A First Step with Orangutan Productions

**DOI:** 10.3390/ani12202761

**Published:** 2022-10-14

**Authors:** Benjamin Beltzung, Marie Pelé, Julien P. Renoult, Masaki Shimada, Cédric Sueur

**Affiliations:** 1IPHC, University of Strasbourg, CNRS, UMR 7178, 67000 Strasbourg, France; 2ANTHROPO-LAB, ETHICS EA 7446, Université Catholique de Lille, 59000 Lille, France; 3CEFE, University of Montpellier, CNRS, EPHE, IRD, 34293 Montpellier, France; 4Department of Animal Sciences, Teikyo University of Science, 2525, Yatsusawa, Uenohara 409-0193, Yamanashi, Japan; 5University Institute of France, 75231 Paris, France

**Keywords:** primates, deep learning, drawing behavior, artificial intelligence, cognition

## Abstract

**Simple Summary:**

Understanding drawing features is a complex task, particularly concerning non-human primates, where the relevant features may not be the same as those for humans. Here, we propose a methodology for objectively analyzing drawings. To do so, we used deep learning, which allows for automated feature selection and extraction, to classify a female orangutan’s drawings according to the seasons they were produced. We found evidence of seasonal variation in her drawing behavior according to the extracted features, and our results support previous findings that features linked to colors can partly explain seasonal variation. Using grayscale images, we demonstrate that not only do colors contain relevant information but also the shape of the drawings. In addition, this study demonstrates that both the style and content of drawings partly explain seasonal variations.

**Abstract:**

Drawings have been widely used as a window to the mind; as such, they can reveal some aspects of the cognitive and emotional worlds of other animals that can produce them. The study of non-human drawings, however, is limited by human perception, which can bias the methodology and interpretation of the results. Artificial intelligence can circumvent this issue by allowing automated, objective selection of features used to analyze drawings. In this study, we use artificial intelligence to investigate seasonal variations in drawings made by Molly, a female orangutan who produced more than 1299 drawings between 2006 and 2011 at the Tama Zoological Park in Japan. We train the VGG19 model to first classify the drawings according to the season in which they are produced. The results show that deep learning is able to identify subtle but significant seasonal variations in Molly’s drawings, with a classification accuracy of 41.6%. We use VGG19 to investigate the features that influence this seasonal variation. We analyze separate features, both simple and complex, related to color and patterning, and to drawing content and style. Content and style classification show maximum performance for moderately complex, highly complex, and holistic features, respectively. We also show that both color and patterning drive seasonal variation, with the latter being more important than the former. This study demonstrates how deep learning can be used to objectively analyze non-figurative drawings and calls for applications to non-primate species and scribbles made by human toddlers.

## 1. Introduction

Anthropocentric bias can arise when interpreting the meaning of what is depicted in drawings. Most studies on figurative drawings; nevertheless, non-figurative drawings can also have meanings, as found in young children’s drawings [1]. Moreover, drawings contain a large amount of information, and using a predefined set of handcrafted features limits the amount of information that can be extracted. This is particularly true when studying drawings made by non-human primates, as there may be an anthropocentric bias in choosing those features that are expected to be meaningful to us, even though they may be totally irrelevant to other species. Thus, predefined handcrafted features do not fully exploit the information content of drawings. In humans, a possible way to mitigate this issue is to ask the drawer what he or she intends to represent [2]. However, the drawer may not be directly aware of the deep meaning of the produced drawing. For example, the method cannot be applied to scribbles drawn by very young children who are not able to communicate verbally or to children who cannot communicate because of pathologies. Obviously, verbal communication about intentions is also not applicable to non-human animals that can draw, such as other primate species [3].

In the present study, we used artificial intelligence to analyze the drawings of a female orangutan named Molly († in 2011), who produced 1299 drawings in her last five years as part of a behavioral enrichment program [4] at the Tama Zoo in Japan. Previous investigations of Molly’s drawings have already demonstrated that her drawings were influenced by her keeper’s identity and daily external events, such as the birth of conspecifics [4]. The authors of [3] investigated the differences among Molly’s drawings using traditional ethological methodologies (i.e., by selecting features and manually extracting them). These authors demonstrated the evolution of Molly’s drawings over time and the influence of seasons. For instance, Molly used green color mostly during the summer and winter seasons and pink color in spring and autumn. How she drew lines also differed according to the season. However, because of the time required to extract features manually, these analyses were conducted on only 749 of the 1299 available drawings. Moreover, the authors focused on features typically used to study children’s drawings [5], such as loops, circles, and fan patterns, which may have a different meaning for orangutans and humans or may even be totally irrelevant.

Deep learning can overcome these limitations. Deep learning is a set of artificial intelligence methods based on artificial neural networks [6,7] and is now widely used in computer vision. Deep learning is an efficient way to replace or complement human expertise in images analyzed in complex tasks, such as classifying microscopy images [8] and diagnosing pathologies [9]. Deep learning does not require handcrafted features; it only requires labeled images. Convolutional neural networks (CNNs), the most common class of deep learning algorithms in image analysis, automatically learn and extract features that are most relevant for a given task (e.g., classification of images based on their labels). CNNs are composed of multiple layers, each containing filters, which are matrices that learn feature representations. The filters are initially randomly initialized and then optimized during the training. The complexity of the features increases with the depth of the layers, from low-level features (e.g., edges, circles) in shallow layers (i.e., close to the input) to high-level features (e.g., a nose, eyes) and entire objects (e.g., a face) in deeper layers (i.e., close to the output). Convoluting a filter with an image results in a new matrix, called a feature map. A feature map, also known as an activation map, represents the activation of the filter in different parts of the image. A high activation indicates the occurrence of a feature encoded by the filter in this part of the image. CNNs allow for high accuracy but have notoriously low interpretability. Nevertheless, multiple methods have recently been developed to better understand deep learning models, such as visualizing which parts of an image are important for its classification [10].

Despite the power of artificial intelligence to analyze images, this method has rarely been applied to understand drawing behavior [11]. Thus far, most applications have exploited the high performance of deep learning in classification, for example, to recognize stroke representations [12] or to classify entire drawings according to the depicted object category [13]. However, the use of deep learning to investigate ontogeny or phylogeny has received little attention.

Here, we investigate multiple methods of using deep learning to decipher Molly’s drawings. We first tested whether season could explain the variation among Molly’s drawings by training a deep learning model to classify each drawing into one of the four seasons during which it was created. The results confirm the seasonal effect revealed in a previous study using handcrafted features [3]. To further understand this effect, we analyze how the information spreads within the network from shallow to deep layers. More precisely, we investigated the impact of feature complexity on classification accuracy. Finally, we leverage a particular type of artificial intelligence technique named ‘style transfer’ to dissociate features associated with drawing style from those describing the representational content, and to analyze the relative importance of these two types of features in explaining seasonal variation in the predictions of the model.

## 2. Materials and Methods

### 2.1. Dataset

The data consist of 1299 drawings realized between 2006 and 2011 by Molly, who started drawing at approximately the age of 50 years and rarely had contact with conspecifics. She spent the morning in an enclosure (that could be inside or outside), and the afternoon in a restroom, where a crayon bucket was at her disposal so she could draw freely and whenever she wanted to. She spent approximately from 2 to 3 h every day, at the rate of 1–2 drawings per day. To end a session, Molly placed the drawing supplies on the floor. For more details on Molly and her drawings, please refer to [3,4]. During this period, she was provided a piece of paperboard and colored crayons daily. As this activity was not specifically meant to investigate her drawing behavior, no rigorous protocol was applied. Thus, only few metadata were collected. To the best of our knowledge, the date of the production of the drawings is the only external source of information available. Each drawing was then categorized into one of the following four seasons based on the date: autumn, summer, spring, and winter (374, 284, 269, and 372 drawings, respectively). To fit the model discussed below, the drawings were reshaped to squares and resized to (224, 224, and 3). The dataset was split into a training set and a validation set with 907 and 392 images, respectively. For each set and season, the drawings were distributed as follows: 28.8% in autumn, 20.7% in spring, 21.9% in summer, and 28.6% in winter. Examples of drawings for each season are shown in Figure 1.

Apart from Molly’s drawings, we considered two other datasets of animal drawings that should be easy to classify. We used these datasets to perform a sanity check on the performance of our artificial intelligence model. Pigcasso’s dataset [14] includes 170 drawings of a male pig (*Sus scrofa*) living in South Africa who paints on a canvas by holding a brush in his snout. Pigcasso’s drawing behavior was enhanced by positive reinforcement. The Pockets Warhol dataset [15] consists of 170 drawings made by a capuchin monkey (*Cebus capucinus*), who draws in a sanctuary in Canada by putting paint on his fingers and applying it to a canvas. Pockets Warhol started drawing as he was given children’s paints to occupy himself.

### 2.2. Convolutional Neural Network (CNN)

As features pre-selected by humans may not grasp all the relevant information in drawings, we focused on deep learning, and more precisely, on CNN. The first CNN was trained on Molly’s drawings to predict the season. We used the transfer learning method [16] to circumvent the limitations of our relatively small dataset. Transfer learning consists of using an existing model that has already been trained on a particular task to perform another task. Following previous studies on drawings [17,18], we used the VGG19 architecture with ImageNet weights. VGG19 consists of 16 convolutional layers followed by 3 fully connected (FC) layers. We transferred the weights of the convolutional layers and randomly initialized the weights of the FC layers. The number of channels starts from 64 in the early layers, which captures simple elements such as lines, curves, and blobs [19], and grows to 512 in the deepest convolutional layers, allowing the network to learn a representation of complex objects. All convolution kernels were 3 × 3. We tested different parameters and hyperparameters (e.g., learning rate and length of FC layers) to find the best model, the architecture of which is described as follows. The last convolutional layer from VGG19 is followed by an average pooling layer and a batch normalization layer. Then, a first FC layer of size 2048 with L2 regularization (strength of regularization of 0.1) is added, as well as a dropout layer with a rate of 0.2. The last layer is an FC layer of size 4, and each neuron corresponds to a season with softmax activation. We use a categorical cross-entropy loss function. To train the model, we used a stochastic gradient descent (SGD) optimizer with a learning rate of 0.1, and a batch size of 16. The best model was determined through early stopping by monitoring the validation set accuracy, and the training was stopped after three epochs with no improvement. As VGG19 is not specifically trained on drawings or paintings, we froze all convolutional blocks. To increase model accuracy, we also used data augmentation (horizontal and vertical flips). Illustration of the VGG19 architecture and examples of feature maps are shown in Figure 2.

### 2.3. Activations Classification

The previous classification of drawings according to season based on transfer learning was used as a baseline to compare other classification approaches intended to explore the role of different features in seasonal variation. Here, we focus on the features described by activations [20]. As shallow layers encode local information represented by simple features, and deeper layers encode global and more complex information, we first compared the ability of these different layers to classify seasons. For this purpose, each drawing was encoded using VGG19 (pretrained with ImageNet weights [21]), and for each layer separately, its activations were extracted, reduced, and used for classification. For the first convolutional layer, for example, one image is represented by 64 matrices f_1_, …, f_64_ of size 224 × 224, corresponding to the 64 feature maps f_i_ of this layer. The maps are then flattened and concatenated into a single activation vector *f*. Then, principal component analysis is applied to the matrix containing the activation vector of every image, and only the first principal components (PCs) explaining a total of 80% of the variance are retained. The new vectors containing the retained PCs scores are then fed to a support vector machine (SVM) to classify images according to season. The same procedure is repeated for each convolutional layer, and for the first two fully connected layers, except that the activations were already vectorized, and thus did not require flattening. Accuracy was calculated through 10-fold cross-validation. Cross-validation consists of splitting the data into *k*-folds (here, *k* = 10) as follows: one-fold is used as the evaluation set and the *k-1* other folds are used to train the SVM. This process is repeated for *k* folds; thus, each fold is used once for evaluation. For a given layer, the classification accuracy was calculated as the average of *k* scores.

For a given image, the length of *f* decreases with the depth of the layer. Thus, activation extraction does not require a significant amount of memory in deep layers. However, extracting the activations of every image requires a large amount of memory for early convolutional layers. In the first convolutional layer, *f* contains 3,211,264 activations. This became an issue when computing PCA for the first two convolutional layers. To address this issue, we reduce the size of *f* by randomly sampling 50% of its activation. To check whether this reduction preserved the information originally contained in the layers, we repeated the *f* reduction 50 times and compared the 50 means of the reduced *f* to the mean of the original *f* using a one-sample *t*-test. This test, performed for both the first and second convolutional layers, was not significant, demonstrating that sampling 50% of the activations did not change their mean. Thus, for the first two convolutional layers, we considered half of the activations to compute PCA.

To understand the importance of colors in the seasonal variation of drawings, the previous SVM-based classification procedure was repeated using either RGB or grayscale images as inputs. The grayscale images were obtained using RGB channels and the National Television System Committee (NTSC) standard. To fit the VGG19 input shape, the new grayscale channel was repeated 3 times. For each layer, the RGB model was first compared to a dummy classifier to assess whether the RGB model was better than random classification. The RGB model was then compared with the grayscale model. To do so, we performed a 5 × 2 cv paired *t*-test [22]. This test consists of comparing the accuracy of two models A and B by proceeding to five 2-fold cross-validations in a row, with the null hypothesis that both models have equal performance. The first accuracy is calculated using 50% of the data as training and 50% as testing, before reversing the roles. This was repeated five times for models A and B.

### 2.4. Gram Matrices Classification

Activation maps are often thought to encode information regarding the content of an image. To analyze whether Molly’s drawing style varied across seasons, we repeated the classification procedures using Gram matrices of activations instead of the activations themselves [23]. For a given layer, the feature maps f_i_ are flattened and concatenated vertically to produce matrix *F*. The Gram matrix was then calculated as *G = F^T^F*. Each value of *G* can be interpreted as the correlation between the two feature maps of the image. *G* was computed for every image before flattening. Each resulting vector represents a drawing, and these vectors are fed to an SVM to classify the style of the images according to the season through a 10-fold cross-validation. The classification is repeated for every convolutional layer because the Gram matrices cannot be computed for the FC layers.

All the analyses were performed using Python.

## 3. Results

### 3.1. Classification through Transfer Learning

The model trained to classify drawings according to seasons achieved 41.6% accuracy on the test set. This accuracy is higher than that expected by random (approximately 29%, by always classifying drawings as the most common class). Drawings produced in the autumn (resp. spring) are often predicted to be produced in winter (resp. summer), and inversely, as shown by the confusion matrix (Figure 3). To assess the extent to which these two possible subgroups (autumn and winter vs. spring and summer) may differ, we retrained a model to classify these two subgroups, which achieved an accuracy of 66%.

For comparison, the same model that achieved 41.6% accuracy was trained to distinguish between Molly’s drawings, Pigcasso’s drawings, and Pockets Warhol’s drawings. The model achieved 93.6% accuracy in the 3-category classification task. As an additional comparison, the same model trained to classify Molly’s drawings according to drawing coverage (binary classification: low coverage vs. high coverage) achieved 94.8% accuracy. These results show that our model is effectively able to classify drawings when the differences are large; thus, the significant but relatively low accuracy of the seasonal classification model is due to only subtle differences between seasons in Molly’s drawings.

### 3.2. Layer-Wise Activation-Based and Gram Matrix-Based Classifications

The mean accuracy of the 10-fold cross-validation SVM at each layer for the RGB and grayscale images is shown in Figure 4. The accuracy of classification models based on activations of RGB images was compared to the accuracy of dummy models (always classifying images as belonging to the most frequent class) by conducting a 5 × 2 cv paired *t*-test. For every convolutional and fully connected layer, the model accuracy was different (*p* < 0.05) from the accuracy of the dummy models, showing that every layer encoded sufficient discriminative information to classify Molly’s drawings according to the season better than random. The slope of the linear regression model predicting the accuracy as a function of layer depth was not significantly different from zero (*p* = 0.11), showing that the accuracy did not improve through layers. Considering grayscale images, the accuracy of the models differed (*p* < 0.05) from that of the dummy model, except for convolutional layers 1, 2, and 11. The slope for the grayscale images was positive and significantly different from zero (*p* < 0.01), indicating that the discriminative information contained in high-level features is not linked to colors only. The same test was performed to compare the accuracy between RGB and grayscale images (rather than the accuracy of dummy models) and revealed statistical differences only for layer FC2.

The results of the Gram matrix-based classifications computed on RGB images are shown in Figure 5. Classification accuracy increased with layer depth (*p* < 0.001), indicating that complex-style features vary across seasons.

## 4. Discussion

The analysis of drawings is usually based on a limited number of features, which may fail to capture all the relevant information conveyed by this rich and generally meaningful communication and artistic medium. Moreover, the study of drawings is also limited by our subjective, human-based interpretation and understanding of what is represented, particularly for non-representative drawings, such as those produced by non-human primates. In this study, we proposed the use of deep learning models to overcome the limitations of traditional drawing analyses, as feature selection and extraction are not performed manually. Here, we focused on the classification of drawings of one female orangutan according to the season in which they were drawn. Previous analyses have shown differences based on the main color, where the main color is green in summer and winter, and pink in spring and autumn [3]. Thus, this study aimed to use deep learning to analyze these drawings and interpret the differences found in seasons and compare our results with previous findings. To do so, we focused on convolutional neural networks (CNNs), the most common class of deep-learning models. By comparing the model accuracy with random predictions, we showed that the trained model was able to correctly classify drawings according to the season of production. Then, by exploring the information encoded by different levels of deep learning models, we could reveal how color, representational content, drawing style, and feature complexity contribute to seasonal variation in drawings.

Our general finding that deep learning methods can successfully encode visual information and detect variations among drawings is consistent with that of previous studies on abstract paintings. For example, [24] used a neural network to detect the correct orientation in such paintings, and [25] used deep learning to recognize art styles in paintings, notably, abstract art. Our results extend these reports by showing that deep learning can discover structured variations that have often been considered scribbles with unpredictable variations in the literature [26].

Although the CNN model classification was better than random classification, the achieved accuracy remained relatively low (41.6%). This result could be explained by multiple reasons. The first interpretation comes from the confusion matrix, which shows that drawings produced in autumn are mostly misclassified as winter; conversely, drawings produced in winter are mostly misclassified as autumn. The same pattern was observed in spring and summer. These results are inconsistent with those of [3], in which the suggested subgroups were summer and winter in autumn and spring. This may be because the CNN analyzes more complex features that were not considered in [3]. The relatively low accuracy of 41.6% may also be explained by the lack of a protocol when the drawings were realized, but also by external events that occurred at Tama Zoological Park. For example, [4] observed that the drawing produced by Molly on the day another orangutan gave birth was particularly red. They also observed that when elementary school students visited the zoo, the drawings were particularly rich in color and lines. It has also been demonstrated that the behavior of captive orangutans is directly linked to their familiarity with the keeper [27]. The sex of the keeper and the people Molly has met could also influence her behavior, as was found for a female Japanese monkey [28], whose arterial blood pressure was different in the presence of men and women, and of strangers and caretakers. All these environmental factors likely added heterogeneity to Molly’s drawings, which contributed to blurring the seasonal variation.

To further explore this seasonal variation, we considered that an image can be seen as an association of the following two components: content and style, and that these two components can be analyzed separately. Feature maps are commonly associated with the content of the image, where the content is ‘the things represented or suggested in something written or created as art, or the ideas it communicates’, according to the Cambridge Dictionary [29]. For a given image, the content will be encoded through all the layers and can be analyzed by extracting activations. The complexity of the content grasped in the activations increases with layer depth. The classification of the extracted activations at each layer was significantly better than random classification, showing the latent complexity of Molly’s drawings, in which activations (i.e., content features) from low to high levels can be discriminated to a certain extent. The accuracy of activation extractions tends to increase from shallow layers to mid-layers, demonstrating the need for more complex and global content features than simple edges. However, the slope of the accuracy as a function of layer depth was not significantly positive, demonstrating that high-level features linked to the content, describing the drawing as a whole, do not contain more relevant information for prediction than mid-level content features. However, this was not the case for grayscale images, where the slope was significantly different from zero. This difference between the analyses of color and grayscale versions of the drawings demonstrates that color information on seasonal variation in Molly’s drawings is encoded more in the early layers than in the deeper layers. This is not surprising, considering that traditional analyses provide evidence for a difference in main colors depending on the season [3]. Indeed, the main color is a low-level feature, which should be captured in the early layers, which could partly explain the importance of colors in the early layers. It is important to note that, even if not significant, the accuracy of the RGB models seems higher than that of the grayscale models. This slight difference may be explained by the features linked to colors that contain information to predict the season of the drawings. Moreover, [30] demonstrated that shallow layers in VGG models encode information on the average hue and other low-level features, saturation, and other variables, whereas high-level features (i.e., deeper layers) can be linked to the concreteness and dynamics of the drawing. It is a complex task to disentangle high-level features that could have played a role in the predictions, but the average hue could be a proxy for the main color that was already analyzed in Molly’s drawings [3]. The accuracies of the RGB and grayscale models were significantly different only for layer FC2. With our analyses, it is not possible to determine whether the relevant features for the RGB and grayscale models are the same. Moreover, differences between seasons are not only related to colors, as grayscale models are better than random models. As RGB models are not significantly better than grayscale models (except for FC2), adding colors only slightly refines these differences, showing that the seasonal variation lies more in the shape of the strokes than their colors.

To complement the study of Molly’s drawings, which describes what is depicted, we further investigated Molly’s drawing style. To do so, we extracted Gram matrices that have been widely used in deep learning. However, how Gram matrices and styles are linked is not self-explanatory. Gram matrices in neural networks were first proposed to analyze the texture of images [23]. Here, the word ’style also refers to the ‘texture’. No consensus has been established on the meaning of these two words; however, according to [31], the region in an image has a constant texture if a set of local statistics or other local properties of the picture function are constant, slowly varying, or approximately periodic. The introduction of Gram matrices can be applied to analyze the style of an image dating back to [32]. In their study, the authors proposed decomposing artistic images into content and style, with the goal of transferring the style of an image onto another image whose content is kept unchanged. Our results suggest that the content represented by neuronal activation is not sufficient to extract all the latent information conveyed by Molly’s drawings. First, the accuracy based on Gram matrices was higher than random for every layer, showing that Molly’s drawing style varied across seasons. More importantly, style-based classification became more accurate when information from deeper layers was considered. This demonstrates that the layout of the drawings, that is, their global stylistic aspect, prevails over the local stylistic features. While the peak accuracy is reached in the intermediate layers in content analyses, it is reached in the deepest layers for style, which highlights the importance of considering aspects, content, and style in drawing analysis.

The PCA applied to the Gram matrices showed that 80% of the variation could be retained with only two principal components for the first two convolutional layers. The corresponding scatter plots of PC1 versus PC2 revealed two possible clusters that did not appear to be season-dependent (Figure 6). With current data, it is difficult to identify the factors that explain these two clusters. They may be linked to the identity of the keeper or to various events that occurred at the Tama Zoological Park [4]. However, the corresponding scatter plots obtained from the activation vector (i.e., for content-based classification) also exhibited two clusters. These clusters become less distinct and less marked when looking at deeper layers and completely vanish in the deepest layers, suggesting that they are primarily rooted in low-level features such as edges (orientation and width).

The approach of this study is based on the fact that biases can arise in analyses that require manual feature selection and extraction. However, deep learning reproduces biases such as race biases [33]. Because we used transfer learning through the VGG19 architecture pretrained on a global classification task (the ImageNet dataset), the features learned by the model are generic. The choice of architecture is not crucial, as it has been shown that features from ImageNet pretrained VGG19 are shared when using other architectures [34].

Analyzing activations and Gram matrices does not require feature selection and extraction, thus being an objective methodology. Of course, the approach proposed here is still slightly anthropocentric when interpreting the results. Indeed, the concepts of content and style are human ones, and Molly may not perceive these concepts the same way as humans do. Deep learning and feature extraction partially allow for getting rid of anthropocentric biases as no feature selection or extraction involves humans. The authors of [3] extracted 12 simple and objective features from orangutan drawings (e.g., coverage rate, distance to the center, number of loops, number of triangles), including Molly’s ones, before performing a PCA, and the features the most correlated with the first axis were also highly linked to the content of the drawings whilst variables in the second correspond more to the style, demonstrating that even objective features will be linked to subjective ones. Whatever the approach, some anthropocentric bias linked to the production of the drawings will always remain. For example, the perceived affordance of a white canvas will be different for a human and a chimpanzee, and there will also be variability between individuals [35]. Regardless of these biases linked to the models or data collection procedures, the interpretation of the results will be performed by humans, making it impossible to completely erase these anthropocentric biases. As Nagel wrote [36], a human could never be able to know what it is to be another animal, but we think that artificial intelligence can help to decrease the number of biases.

## 5. Conclusions

Deep learning has great potential for improving our understanding of drawing behavior, both ontogenetically and phylogenetically. Such models can grasp the latent features present in abstract drawings, which is a time-consuming, laborious, and complex task for traditional analyses. The present case study on Molly’s drawings demonstrates how the complexity of content and style, which is directly linked to Molly’s drawing behavior, can be studied through a CNN, from shallow to deep layers. The layers of a CNN can be seen as the layers of a painting as follows: each layer contains partial information about the art at different levels of detail (such as colors, shapes, and contrast), and to understand a painting, one needs to combine these layers.

This study brings new advances in the understanding of drawings in non-human primates and can give highlights both on fundamental and more applied research. As regards non-verbal drawers, applying the current deep learning methodology to several ape species may lead to discoveries in the domain of evolutionary anthropology and comparative psychology. Moreover, analyzing drawings thanks to deep learning may help to assess neurodegenerative diseases in apes and enhance animal welfare. In humans, problems with languages (forgetting words, persons, etc.) allow targeting such problems but this cannot be performed with apes. Changes identified by AI in drawings with the age of individuals may help to identify such diseases.

## Figures and Tables

**Figure 1 animals-12-02761-f001:**
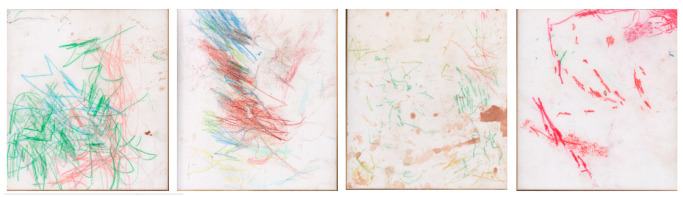
Examples of Molly’s drawings. The season of each drawing, from the left to the right, is autumn, spring, summer, and winter. Credits goes to the Tama Zoological Park.

**Figure 2 animals-12-02761-f002:**
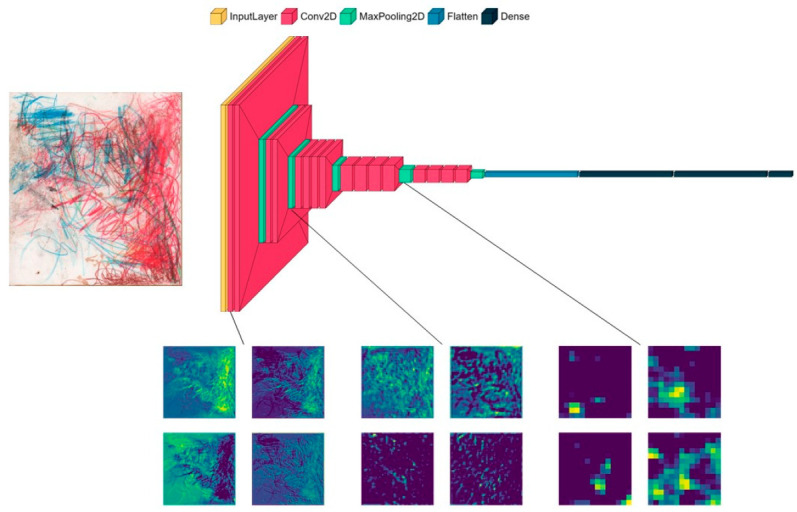
VGG19 model architecture and examples of feature maps (on the bottom) for a drawing. From left to right, each 2 × 2 square of feature maps are extracted from block1_conv1, block3_conv1, block5_conv1, respectively. The early blocks are in a high resolution, as the filters capture fine details, compared to deeper blocks, where the filters detect more general concepts.

**Figure 3 animals-12-02761-f003:**
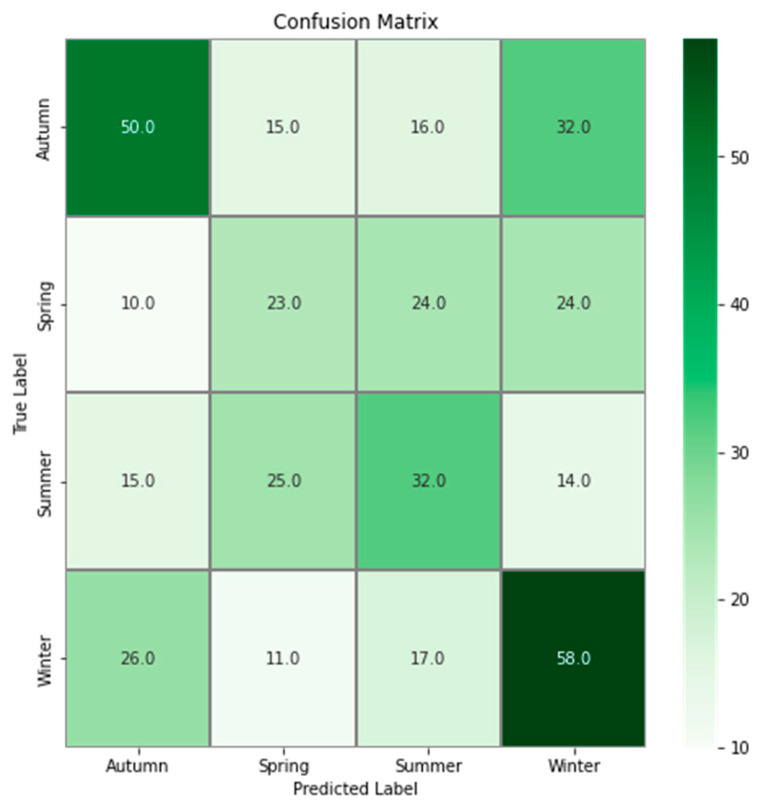
Confusion matrix of classification between seasons. For a given element, the value corresponds to the number of drawings predicted as the label matching with the column but belonging to the label matching with the row.

**Figure 4 animals-12-02761-f004:**
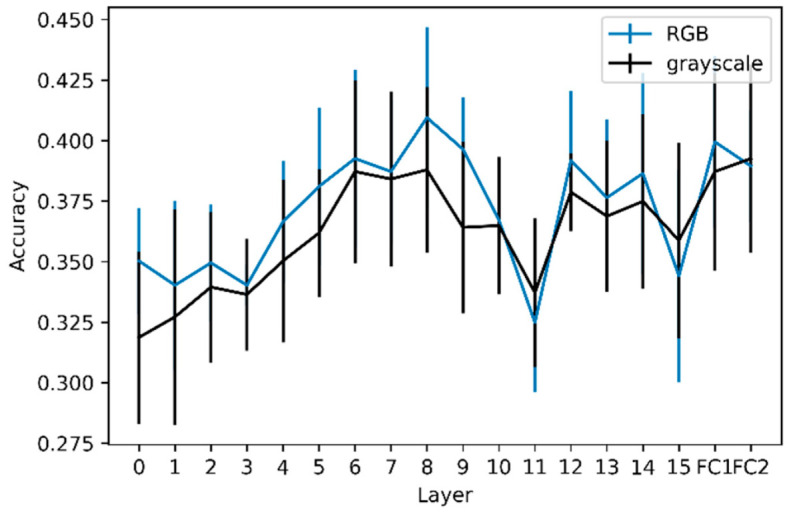
Mean accuracy of the 10-fold cross-validation for RGB images and grayscale images for each layer. The integer values on the x axis represent the convolutional layer on which the accuracy has been calculated, while FC1 and FC2, respectively, represent the first and second fully connected layers. The vertical bars represent the standard deviations.

**Figure 5 animals-12-02761-f005:**
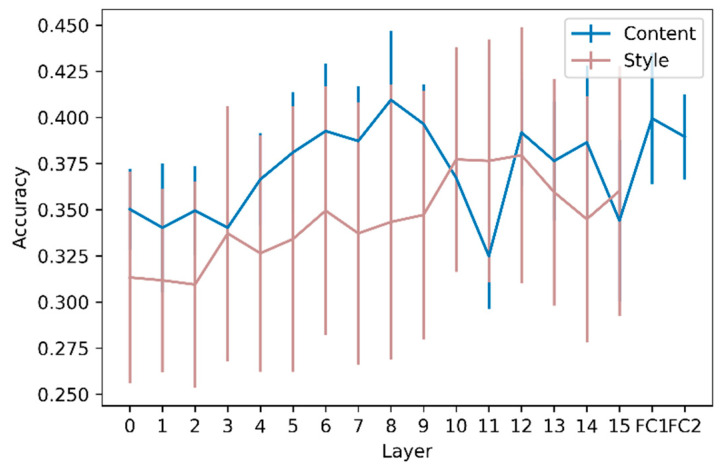
Accuracy of the models based on the Gram matrices (in brown) and on the RGB feature maps (in blue) for each layer. The vertical bars represent the standard deviations. The highest accuracy for Gram matrices classification is 37.9% for the 13th layer.

**Figure 6 animals-12-02761-f006:**
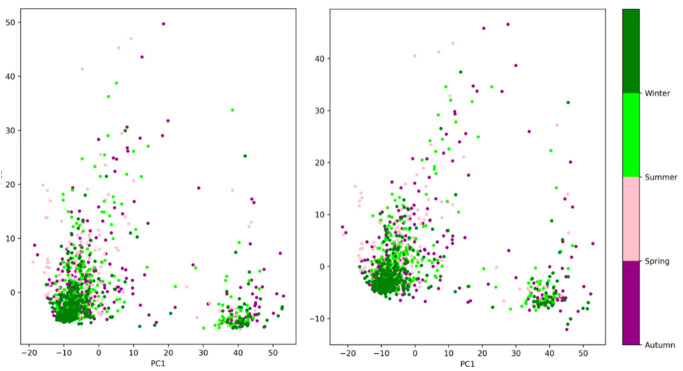
Scatter plot of the two PC of the Gram matrices of the first convolutional layer (on the **left**) and of the second (on the **right**), where each point is a drawing.

## Data Availability

The data presented in this study are available on request from the corresponding author. The data are not publicly available due to Tama Zoological Park’s request and copyright.

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
