# Peer review of "Using Artificial Intelligence to Analyze Non-Human Drawings: A First Step with Orangutan Productions"

_animals, 2022, doi:10.3390/ani12202761_

Round 1

Reviewer 1 Report

This is a fascinating and well written piece of work. I have little to say regarding changes/improvements and would be happy to see it published in its current form (after a few minor language/sentence structure edits).

As an interested non-expert in the field of machine learning the MS might benefit from a few visual representations of what the VGG19 model is doing throughout the layered classification process. Overall, I think the methods are explained clearly and with sufficient detail, particularly given their complexity and potential unfamiliarity to readers, but a few visual cues might aid comprehension.

The only other comment I have is that it might broaden the appeal of the MS if the implications and applications of this technique were discussed a little more thoroughly, even if speculatively. For example, are there practical applications of this approach (evaluating captive animal enrichment e.g.?) beyond the already valid and valuable presentation of the method?

Reviewer 2 Report

Beltzung and colleagues present a well-written and comprehensive study in which they apply artificial intelligence methodologies to analyze a large collection of drawings made by an adult female orangutan. Their results showed that their deep learning model was superior to random guessing in determining from the aesthetic features of the drawing the season that it was created. Their method, in contrast to traditional subjective interpretations of non-human animal artwork, takes a quantitative approach and relies on machine-based detection of features and the identification of their complexity. The method is novel and innovative, and it offers to provide insights into the drawing behavior of not only great apes but also potentially other non-human animals and also perhaps non-verbal humans. I believe the paper merits acceptance in Animals so long as the items outlined below are addressed in a minor revision.

Major Comments

Line 110. The circumstances under which the drawings were created could be better explained. Was Molly separated from other orangutans when the drawings were created? How did the drawing activity fit within her daily routine?  Was the orangutan on exhibit or in the holding area? What was the level of outdoor-access during each season? Also, the methods should ideally include more details about the drawing task itself: were all of the crayons available to the ape so that they decided when to start/stop with crayon each on their own, or was a human keeper/experimenter involved in facilitating the handover of individual crayons? Did all the sessions typically last the same amount of time? How was it determined (by the ape or by a human) when a drawing sessions would end and a drawing was complete? Please provide these details.

Line 334: On the one hand, the authors contend that traditional methods for analyzing non-human art likely contain anthropocentric bias because they involve subjective judgement about which features are meaningful (line 47), yet on the other hand, it seems to me that the authors show a similar bias by imposing human artistic standards for “content and style” onto the orangutan’s drawings. They do this by conflating the idea of “content” (which they define as “things represented or suggested in something written or created as art”) with the complexity of extracted activations at different layers of feature maps in the orangutans drawings. It seems to me that according to the provided definition of “content”, the crucial aspect is not the complexity of the scribbled forms per se, but rather that they are representational of something else, i.e. symbolic or semiotic in nature.  Without sufficient evidence that Molly, beyond making complex shapes, was also drawing them with the intent of making them representational or suggestive of things, I’m not convinced its appropriate to apply the human-centric concepts of “content and style” to her artwork. With this in mind I encourage the authors to reframe the way in which they discuss their complexity analysis.

Minor Comments

139: Should “drawing as he was given” be changed to “drawing as soon as he was given”?
